# SCHUR-A*: Layer-wise Optimal Expert Pruning for MoEs via Schur-Complement Guided A* Search

**Zheng Chen** [1,2] **Weifeng Yang** [1] **Jianxiao Tang** [1] **Buhui Yao** [1]

## Abstract

Sparse Mixture-of-Experts (MoE) language models enable conditional computation but face deployment challenges due to the *memory wall*: while few experts are activated per token, the entire model must reside in memory. Existing expert pruning methods primarily rely on independent ranking, failing to account for the complex interdependencies and redundancies between experts. In this paper, we formulate post-training MoE pruning as a reconstruction-driven subset selection problem, aiming to minimize layer-output distortion under a cardinality constraint. We introduce SCHUR-A*, an algorithm that leverages A* search to achieve globally optimal expert selection within each layer. To maintain computational tractability, we derive a novel, admissible heuristic upper bound using a Schur-complement-based relaxation of the reconstruction objective. This tight bound allows for aggressive pruning of the search space while mathematically guaranteeing optimality. Furthermore, we propose an automated strategy to balance fidelity and memory reduction across heterogeneous layers via knee-point detection. Extensive experiments on Qwen3-30B-A3B demonstrate that SCHUR-A* significantly outperforms greedy and ranking-based baselines, maintaining comparable performance even under aggressive pruning ratios.

## 1. Introduction

The Mixture-of-Experts (MoE) paradigm has revolutionized LLM scaling by decoupling model capacity from inference cost (Shazeer et al., 2017; Lepikhin et al., 2020). By activating sparse expert subsets, models like Mixtral (Jiang et al., 2024) and DeepSeek-V2 (Shao et al., 2024) achieve high performance while maintaining efficient FLOPs. However, this efficiency is hindered by a *memory wall*: the massive count of *resident* parameters necessitates immense GPU memory, restricting deployment on commodity hardware (Rajbhandari et al., 2022; Yao et al., 2024; Kim et al., 2023; Pope et al., 2022).

To mitigate this, expert pruning aims to remove redundant experts. Traditional methods typically rely on *separable heuristics*, ranking experts by isolated metrics like routing frequency (Xie et al., 2024). As shown in Figure 1, such rankings fail to capture *combinatorial synergies*—where a weak expert may provide niche, indispensable knowledge. Any separable ranking implicitly assumes submodularity, which does not hold under correlated experts. While reconstruction-based objectives (Lu et al., 2024) theoretically address this, they encounter a combinatorial explosion ($\binom{E}{r}$) in modern fine-grained MoEs, forcing a reliance on suboptimal greedy search.

To bridge the gap between optimality and tractability, we propose SCHUR-A*, a rigorous framework that addresses layer-wise pruning as a combinatorial search problem. While greedy search or exhaustive enumeration may suffice for simpler MoE architectures with few experts (e.g., $E = 8$), SCHUR-A* targets the emerging regime of fine-grained MoEs with 64–128+ experts. In these massive search spaces, where greedy search is prone to suboptimal local traps and enumeration is intractable, our method utilizes an A* search algorithm guided by a spectral heuristic. By leveraging the Schur complement to construct an admissible upper bound, SCHUR-A* effectively prunes suboptimal branches in the massive search space, guaranteeing the discovery of the global optimal expert subset with high efficiency. Our contributions are summarized as follows:

- We reformulate pruning as a combinatorial optimization problem and propose SCHUR-A*, which utilizes a Schur-complement-based bound (Vandenberghe & Boyd, 1996) to efficiently locate the layer-wise optimal expert subset.

[1]School of Information and Software Engineering, University of Electronic Science and Technology of China, Chengdu 610054, China [2]National Key Laboratory on Blind Signal Processing, Chengdu 610041, China. Correspondence to: Zheng Chen <zchen@uestc.edu.cn>.

*Proceedings of the 43rd International Conference on Machine Learning*, Seoul, South Korea. PMLR 306, 2026. Copyright 2026 by the author(s).

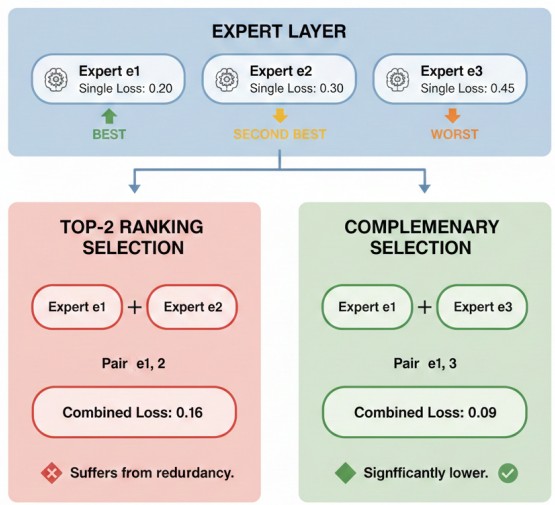

*Figure 1.* An example illustrating why selecting experts based on individual performance (Ranking) is suboptimal. Although Expert $e_3$ has the worst single loss (0.45), it is highly complementary to $e_1$. Consequently, the pair $\{e_1, e_3\}$ achieves a significantly lower combined loss (**0.09**) than the Top-2 ranking selection $\{e_1, e_2\}$ (0.16), which suffers from redundancy.

- We devise a Vectorized Gain Computation scheme via incremental Cholesky updates, effectively reducing the search complexity from cubic to quadratic by simultaneously evaluating all candidates.

- We integrate the Kneedle algorithm (Satopaa et al., 2011) to automatically determine the optimal expert budget (knee-point) for each layer, eliminating the need for manual pruning ratios.

**Conflict of Interest Disclosure** The authors declare no financial conflicts of interest. The models evaluated in this paper are publicly available from open-source repositories.

## 2. Related Work

The evolution of MoE has shifted from foundational sparse gating (Shazeer et al., 2017) and Switch Transformers (Fedus et al., 2021) to high-granularity designs(Pi'oro et al., 2024; Cheng et al., 2025). Mixtral (Jiang et al., 2024) popularized top-2 routing (Zhou et al., 2022)for balanced performance, while DeepSeek-V2 (Shao et al., 2024) further scaled capacity using Multi-head Latent Attention and a vast pool of fine-grained experts. Despite their success, the system-level overhead of hosting these experts remains a primary bottleneck (Gale et al., 2022; Luo et al., 2025; Balmau et al., 2025), motivating our focus on structural compression.

Existing pruning methods primarily prioritize experts using routing statistics or weight magnitudes. *MoE-Pruner* (Xie et al., 2024) focuses on activation frequencies, while Guo

et al. (2025) employs clustering to maintain expert diversity(He et al., 2023). Recent empirical studies (Su et al., 2025)suggest that expert-level pruning preserves model fidelity better than traditional quantization for one-shot compression (Lasby et al., 2025). However, these approaches treat experts as independent entities, failing to optimize for the collective contribution of the selected subset.

Framing pruning (Zhou et al., 2025)as a reconstruction problem—minimizing output distortion—has proven effective for both dense weights and MoE experts (Lu et al., 2024; Frantar et al., 2022). This shift transforms pruning into a discrete combinatorial optimization task. Unlike previous greedy solvers which are prone to local optima in high-dimensional spaces(Das & Kempe, 2011), our work leverages the **Schur complement** (Vandenberghe & Boyd, 1996; Singh & Alistarh, 2020) to navigate the search space. By providing a rigorous upper bound for the A\* algorithm, we achieve the first scalable search that maintains layer-wise optimality for fine-grained architectures.

## 3. Method

In this section, we detail the mathematical formulation and algorithmic design of our framework. We begin by defining the reconstruction objective and the Linear Refitting problem in Section 3.1. Next, we derive the Schur-complement-based heuristic function in Section 3.2, which serves as the theoretical backbone for our global search. We then present the vectorized gain computation scheme that accelerates the solver in Section 3.3. Finally,to eliminate the need for manual ratios, we introduce an adaptive expert budget selection strategy based on the Kneedle algorithm in Section 3.4.

The overall workflow is visually depicted in **Figure 2**. Taking the pre-computed sufficient statistics $(G, u)$ and the determined budget $r_l$ as inputs, the process executes an A\* search to locate the optimal subset $S^*$, followed by a final linear refitting step to output the exact expert weights $\beta^*$.

### 3.1. Objective: Reconstruction with Linear Refitting

We formulate expert pruning as minimizing the layer-wise reconstruction error. To avoid computationally expensive repeated forward passes (Lu et al., 2024), we utilize pre-calculated expert contributions $C_i \in \mathbb{R}^{N \times d}$ (the expert output modulated by its original gate) collected from a single calibration pass. While a naive subset selection minimizes $L(S) = \|Y - \sum_{i \in S} C_i\|_F^2$, this approach fails to address *correlation leakage*—where the information of pruned experts could be recovered by the remaining subset due to non-orthogonality.

To exploit these synergies, we introduce coefficients $\beta$ to perform Linear Refitting, allowing selected experts to lin-

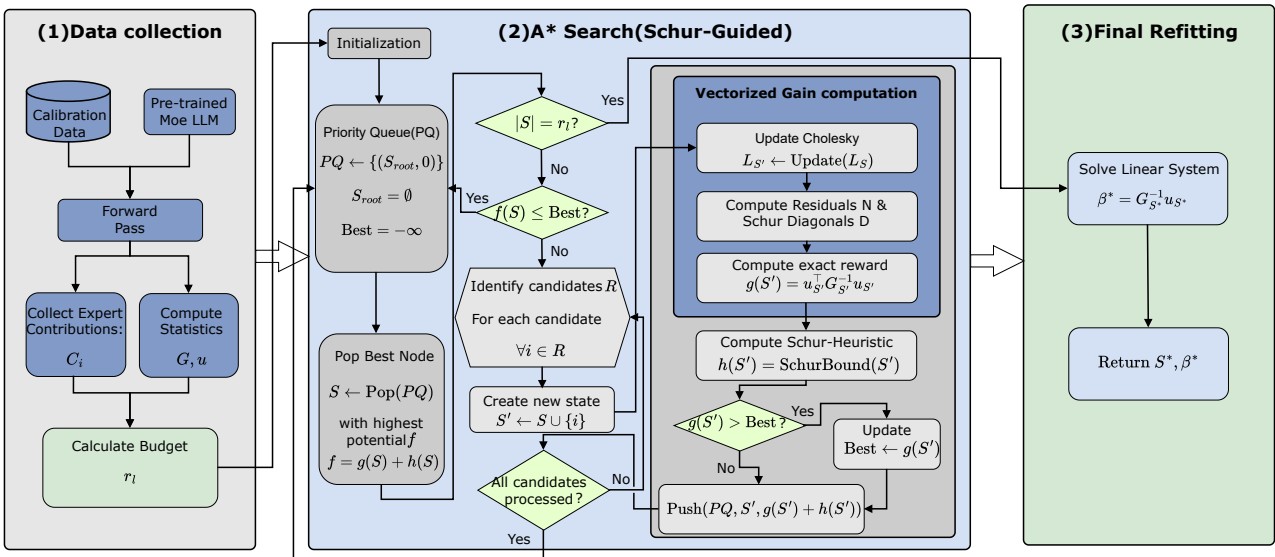

*Figure 2.* **Flowchart of the SCHUR-A* Algorithm.** The workflow consists of three main stages: (1) Data Collection: Pre-computing statistics $(G, u)$ from the calibration set. (2) A* Search: Iteratively expanding the subset $S$ by selecting candidates with the highest potential ($f = g + h$), guided by the exact reward update and Schur-complement heuristic. (3) Final Refitting: Solving the linear system to obtain the optimal coefficients $\beta^*$ for the selected topology.

early compensate for the pruned ones:

$$L^*(S) = \min_\beta \left\| Y - \sum_{i \in S} \beta_i C_i \right\|_F^2 \tag{1}$$

This formulation transforms the task from a greedy search into a basis-selection problem (Nagel et al., 2020). Since the original weights represent just one feasible point in the optimization space of $\beta$, $L^*(S)$ is mathematically guaranteed to be *strictly non-inferior* to fixed-weight selection. Crucially, we utilize the Gram matrix of statistics $C_i$ to solve Eq. 1 efficiently without further model inference.

We distinguish the role of $\beta$ across two phases. In the **Search Phase**, $\beta$ serves as a *selection proxy*, guiding the A* search to the topology $S^*$ with the maximum representational potential. However, in the **Inference Phase**, we treat the usage of $\bar\beta$ as an empirical variable. While $\beta$ minimizes local reconstruction error, it risks overfitting the calibration set. As shown in our experiments, reverting to the robust, pre-trained gating weights for the selected topology $S^*$ consistently yields superior generalization compared to retaining the static refitted coefficients.

### 3.2. Heuristic Function with Schur-Complement Bound

To optimize the linear refitting objective $L^*(S)$ for a fixed subset $S$, we treat it as an Ordinary Least Squares (OLS) problem where setting the partial derivative with respect to $\beta$ to zero yields the analytical solution for the optimal coefficients

$$\beta^* = G_S^{-1} u_S, \tag{2}$$

where $G_S$ is the Gram sub-matrix corresponding to the selected experts $S$, and $u_S$ is the vector of inner products between the target $Y$ and the expert contributions $C_i$. Substituting $\beta^*$ back into the objective function reveals that the minimum reconstruction error is

$$L^*(S) = \|Y\|_F^2 - u_S^\top G_S^{-1} u_S, \tag{3}$$

where $\|Y\|_F^2$ is the total energy of the original MoE output. Since this term remains constant for a given layer, minimizing $L^*(S)$ is mathematically equivalent to maximizing the Reward Function $F(S)$, defined as

$$g(S) = F(S) = u_S^\top G_S^{-1} u_S. \tag{4}$$

The efficiency and optimality of our A* search hinge on an admissible heuristic $h(S, t)$, which must provide a rigorous upper bound on the potential reward from selecting $t = r - |S|$ additional experts from the remaining set $R = \bar{S}$. To derive this bound, we partition the global Gram matrix $G$ and the inner product vector $u$ based on the selected set $S$ and the candidate set $R$ such that

$$G = \begin{bmatrix} G_{SS} & G_{SR} \\ G_{RS} & G_{RR} \end{bmatrix}, \quad u = \begin{bmatrix} u_S \\ u_R \end{bmatrix}. \tag{5}$$

Let $\beta = G_{SS}^{-1} u_S$ be the optimal coefficients for the current set $S$. We characterize the "unexplained" information in the candidate set using two pivotal quantities:

- **Residualized vector** $\tilde{u}_R = u_R - G_{RS}\beta$: This represents the portion of the target $Y$ that cannot be explained by the current experts $S$.

- **Schur complement** $\Sigma_R = G_{RR} - G_{RS}G_{SS}^{-1}G_{SR}$: This matrix captures the self-correlation of the remaining experts after projecting out the components already spanned by $S$.

Crucially, for any future subset $T \subseteq R$, the incremental reward $\Delta F(S; T) = F(S \cup T) - F(S)$ can be decoupled from the current state $S$ via the exact expression

$$\Delta F(S; T) = \tilde{u}_T^\top \Sigma_T^{-1} \tilde{u}_T, \qquad (6)$$

where $\Sigma_T$ is the corresponding principal sub-matrix of $\Sigma_R$. This formulation allows us to bound the maximum possible reward efficiently.

**Admissible Upper Bound.** To ensure admissibility, the heuristic must satisfy $h(S, t) \geq \max_{|T|=t, T \subseteq R} \Delta F(S; T)$. By applying the Rayleigh quotient property $x^\top A^{-1} x \leq \|x\|^2/\lambda_{\min}(A)$ for any positive definite matrix $A \succ 0$, and leveraging the *interlacing property* of eigenvalues (Horn & Johnson, 2012) which implies $1/\lambda_{\min}(\Sigma_T) \leq 1/\lambda_{\min}(\Sigma_R)$, we can bound the gain by

$$\tilde{u}_T^\top \Sigma_T^{-1} \tilde{u}_T \leq \frac{\|\tilde{u}_T\|^2}{\lambda_{\min}(\Sigma_R)}. \qquad (7)$$

Maximizing the right-hand side corresponds to selecting the top $t$ largest elements of $\tilde{u}_i^2$, leading to our admissible heuristic definition

$$h(S, t) = \frac{\sum_{j=1}^t \text{Top}_j\left(\tilde{u}_i^2 : i \in R\right)}{\lambda_{\min}(\Sigma_R) + \epsilon}, \qquad (8)$$

where $\epsilon$ is a small ridge term for numerical stability. This heuristic exploits the full geometry of the remaining experts via $\Sigma_R$, providing a tight yet provably safe upper bound that guarantees the A* search will converge to the layer-wise optimum. A formal proof of admissibility and an empirical analysis of its tightness are provided in Appendix A.1.

### 3.3. Vectorized Gain Computation via Cholesky Update

To handle MoE layers with a large number of experts (e.g., $E \geq 128$), the efficiency of expanding nodes in the A* search tree is paramount. A naive greedy search would independently solve the least squares problem for every candidate expert, incurring a prohibitive computational cost of $\mathcal{O}(m \cdot |S|^3)$, where $m$ is the number of candidates. To overcome this bottleneck, we propose a **Vectorized Gain Computation** scheme that leverages cached Cholesky decompositions to update all candidate gains simultaneously.

For each node $S$ in the search tree, we cache its Cholesky factor $L_S$ such that $G_{SS} = L_S L_S^\top$. This allows us to efficiently compute the current optimal coefficients $\beta_S$ and reward $g(S)$ via triangular solves by solving the systems

$$L_S w = u_S, \quad L_S^\top \beta_S = w, \quad \text{and} \quad g(S) = \|w\|^2. \quad (9)$$

When a new expert is added to $S$, $L_S$ can be updated incrementally via a rank-1 update rather than being recomputed from scratch.

To evaluate the potential candidates $R$, we analyze the augmented reward $F(S \cup \{k\})$. By applying the *block matrix inversion lemma* (Banachiewicz identity), the inverse of the augmented Gram matrix decouples analytically, yielding the exact expression for the marginal gain

$$F(S \cup \{k\}) = \underbrace{u_S^\top G_{SS}^{-1} u_S}_{\text{Current } F(S)} + \underbrace{\frac{(u_k - G_{kS}G_{SS}^{-1}u_S)^2}{G_{kk} - G_{kS}G_{SS}^{-1}G_{Sk}}}_{\text{Marginal Gain } \Delta F_k}. \quad (10)$$

Identifying $\beta_S = G_{SS}^{-1}u_S$, the numerator simplifies to the squared residual correlation $N_k^2 = (u_k - G_{kS}\beta_S)^2$, and the denominator becomes the Schur complement $D_k$, proving that our metric is mathematically exact rather than an approximation.

**Vectorized Implementation and Complexity Reduction.** Directly computing the Schur complement term $G_{kS}G_{SS}^{-1}G_{Sk}$ for each candidate is computationally expensive. We accelerate this by organizing all candidates into a matrix $A = G_{S,R} \in \mathbb{R}^{|S| \times m}$ and computing their projections onto the current subspace simultaneously via a Triangular Solve with Multiple Right-Hand Sides (TRSM), defined as

$$W = L_S^{-1}A, \quad \text{where} \quad L_S L_S^\top = G_{SS}. \quad (11)$$

Using the cached $L_S$, this batched operation requires only $\mathcal{O}(|S|^2 \cdot m)$ FLOPs. With the projection matrix $W$, the Schur complement denominators $D$ and the final gains for all candidates are derived via the efficient element-wise operations

$$D = \text{diag}(G_{RR}) + \epsilon - \text{colsums}(W^2), \quad (12)$$

$$\text{gains} = \frac{(u_R - A^\top \beta_S)^2}{D}. \quad (13)$$

This formulation effectively reduces the dominant complexity from cubic $\mathcal{O}(m \cdot |S|^3)$ to quadratic $\mathcal{O}(|S|^2 \cdot m)$. A visual comparison illustrating this efficiency gain is provided in Appendix Figure 8.Furthermore, the reliance on dense matrix operations makes this approach highly amenable to parallel acceleration on GPU tensor cores.

**Lazy Successor Generation.** Combined with the "Generator" mechanism, we utilize these vectorized gains to identify the top-scoring candidates efficiently, performing the computationally heavier full heuristic calculation $h(S, t)$ only for these promising branches. This hierarchical pruning ensures that the search focuses on the most potential paths without exhausting memory or compute resources.

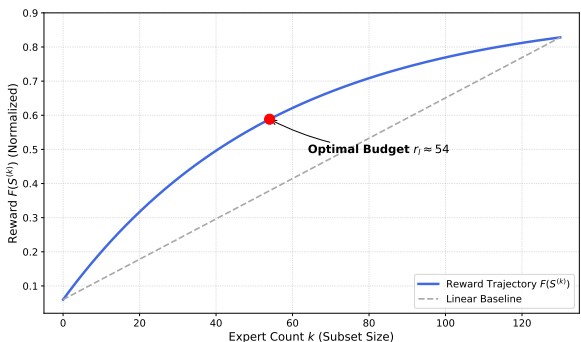

*Figure 3.* **Adaptive Budget Selection via Kneedle.** The optimal budget $r_l$ (red dot) is identified as the point maximizing the vertical deviation $D(k)$ between the normalized reward curve (blue) and the linear baseline (dashed). This captures the onset of diminishing returns.

### 3.4. Expert Budget Selection

Since redundancy varies significantly across layers, uniform pruning ratios are often suboptimal. To address this, we propose an **Adaptive Rank Determination** strategy based on the *Kneedle* algorithm to identify the layer-wise point of diminishing returns.

As shown in Figure 3, we first generate a proxy reward trajectory $\{F(S^{(k)})\}$ using a rapid greedy search (e.g., OMP). Treating this as a continuous curve $f(k)$, we normalize both the expert counts and rewards to $[0, 1]$. The optimal budget $r_l$ is then identified as the "knee" point that maximizes the deviation from the linear baseline:

$$r_l = \arg\max_k \left( f_{norm}(k) - k_{norm} \right) \qquad (14)$$

This maximization of the difference function $D(k)$ mathematically pinpoints the most efficient trade-off between reconstruction fidelity and sparsity. A sensitivity parameter $S$ allows further fine-tuning of this detection strictness.

## 4. Experiments

### 4.1. Experimental Setup

**Models and Benchmarks.** We evaluate our method on the Qwen3-30B-A3B model (Yang et al., 2025) without any fine-tuning, a state-of-the-art fine-grained Mixture-of-Experts LLM. This model comprises approximately 30 billion total parameters with 2.4 billion active parameters per token, employing a fine-grained architecture with $E = 128$ experts per layer and selecting the top-$k = 8$ experts. To assess the impact of pruning on language modeling fidelity, we report perplexity (PPL) on the validation sets of WikiText-2 (Merity et al., 2016) and C4. For downstream task capability, we employ the lm-evaluation-harness framework to evaluate few-shot accuracy across standard benchmarks.

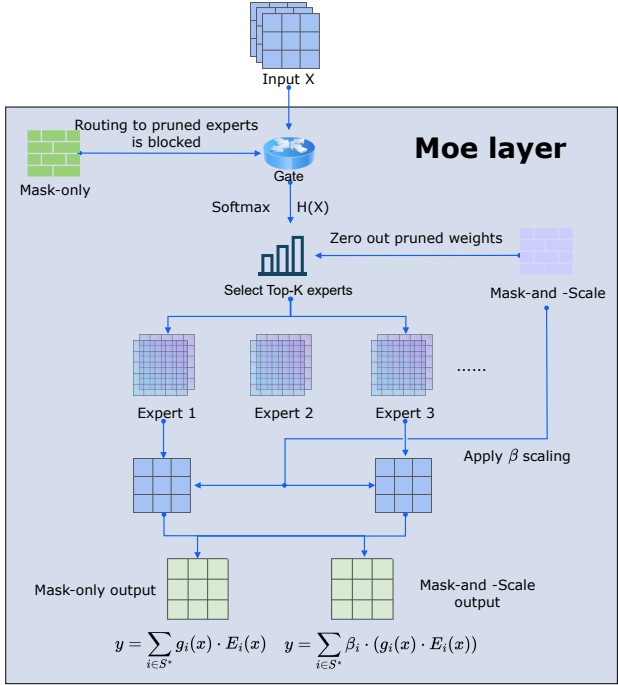

*Figure 4.* **Inference Flow of Pruned MoE Strategies.** The diagram illustrates the operational differences between the two methods: (1) **Mask-Only (Left):** Operates at the gating level. Pruned experts are physically masked (reset to $-\infty$) before the Top-K selection, forcing the router to strictly choose from the retained subset $S^*$ based on original dynamic weights $g(x)$. (2) **Mask-and-Scale (Right):** Operates at the aggregation level. After routing, the outputs of selected experts are modulated by the static refitted coefficients $\beta_i$ (as shown in the formula) to compensate for the information loss from pruned experts.

**Calibration Configuration.** Our pruning is post-training and calibration-based. To construct the calibration dataset $\mathcal{D}_{cal}$, we randomly sample **512 sequences** from the C4 (Colossal Clean Crawled Corpus) dataset (Raffel et al., 2019). Each sequence is truncated to a context length of 2048 tokens to ensure that the collected statistics (Gram matrices $G$ and vectors $u$) cover diverse semantic contexts and positional embeddings. We rely solely on this unlabeled data, ensuring our method remains task-agnostic.

### 4.2. Main Results

We compare SCHUR-A* against three ranking-based baselines: Frequency, Expert Activation Norm (EAN), and REAP (Lasby et al., 2025). To demonstrate the superiority of our combinatorial approach, we evaluate performance across two pruning regimes for the baselines: light pruning (25% pruned) and heavy pruning (50% pruned). Our method operates at an adaptive ratio determined by the Kneedle algorithm, resulting in approximately 42% parameters pruned (57.89% retention).

*Table 1.* **Zero-shot Performance Comparison on Qwen3-30B-A3B.** We compare SCHUR-A* against baselines at 25% and 50% pruning ratios. Despite removing ≈42% of experts, SCHUR-A* (Avg. 0.661) significantly outperforms all 50% baselines (e.g., REAP at 0.518) and achieves performance comparable to the 25% REAP baseline (0.669), effectively bridging the gap between high-sparsity and high-fidelity regimes.

| Method | ARC-c | ARC-e | BoolQ | HellaS | MMLU | OBQA | RTE | WinoG | Avg. |
|---|---|---|---|---|---|---|---|---|---|
| Unpruned Baseline | 0.563 | 0.790 | 0.887 | 0.778 | 0.779 | 0.454 | 0.816 | 0.702 | 0.721 |
| *Baselines at 25% Pruning Ratio (High Retention) (Lasby et al., 2025)* | | | | | | | | | |
| Frequency | $0.401 \pm 0.011$ | $0.600 \pm 0.016$ | $0.847 \pm 0.003$ | $0.593 \pm 0.005$ | $0.600 \pm 0.004$ | $0.342 \pm 0.012$ | $0.781 \pm 0.002$ | $0.637 \pm 0.005$ | $0.600 \pm 0.005$ |
| EAN | $0.406 \pm 0.007$ | $0.603 \pm 0.014$ | $0.847 \pm 0.005$ | $0.607 \pm 0.006$ | $0.600 \pm 0.002$ | $0.337 \pm 0.003$ | $0.764 \pm 0.002$ | $0.660 \pm 0.009$ | $0.603 \pm 0.004$ |
| REAP | $0.481 \pm 0.005$ | $0.720 \pm 0.005$ | $0.852 \pm 0.003$ | $0.706 \pm 0.006$ | $0.674 \pm 0.002$ | $0.405 \pm 0.005$ | $0.813 \pm 0.006$ | $0.701 \pm 0.008$ | $0.669 \pm 0.003$ |
| *Baselines at 50% Pruning Ratio (High Sparsity) (Lasby et al., 2025)* | | | | | | | | | |
| Frequency | $0.285 \pm 0.001$ | $0.424 \pm 0.002$ | $0.779 \pm 0.003$ | $0.458 \pm 0.003$ | $0.397 \pm 0.002$ | $0.286 \pm 0.004$ | $0.659 \pm 0.012$ | $0.570 \pm 0.009$ | $0.483 \pm 0.001$ |
| EAN | $0.296 \pm 0.006$ | $0.426 \pm 0.009$ | $0.759 \pm 0.007$ | $0.471 \pm 0.002$ | $0.443 \pm 0.001$ | $0.291 \pm 0.009$ | $0.668 \pm 0.020$ | $0.589 \pm 0.009$ | $0.493 \pm 0.003$ |
| REAP | $0.344 \pm 0.004$ | $0.504 \pm 0.008$ | $0.745 \pm 0.005$ | $0.489 \pm 0.013$ | $0.507 \pm 0.005$ | $0.311 \pm 0.003$ | $0.625 \pm 0.031$ | $0.623 \pm 0.007$ | $0.518 \pm 0.004$ |
| SCHUR-A* (Ours) | $0.491 \pm 0.014$ | $0.662 \pm 0.010$ | $0.866 \pm 0.006$ | $0.760 \pm 0.004$ | $0.623 \pm 0.004$ | $0.452 \pm 0.022$ | $0.740 \pm 0.026$ | $0.699 \pm 0.013$ | $0.661 \pm 0.005$ |

As shown in Table 1, ranking-based heuristics suffer from catastrophic degradation when sparsity increases. For instance, increasing the pruning ratio from 25% to 50% causes the average accuracy of the strongest baseline, REAP, to plummet from 66.9% to 51.8%. This sharp decline suggests that separable rankings fail to distinguish between truly redundant experts and those that are individually weak but structurally essential. In stark contrast, SCHUR-A* effectively navigates this high-compression regime. Despite pruning ≈42% of experts—closer to the heavy pruning setting—our method achieves an average score of **66.1%**, outperforming the 50% REAP baseline by a massive margin of **+14.3%**. This empirically validates that our Schur-complement-based heuristic successfully preserves the critical "long-tail" experts that greedy methods discard.

The advantage of SCHUR-A* is particularly pronounced in knowledge-intensive and reasoning tasks, which typically rely on the synergistic activation of multiple experts. On the challenging MMLU benchmark, SCHUR-A* achieves an accuracy of 62.3%, significantly surpassing the 50% REAP baseline (50.7%) and approaching the 25% REAP baseline (67.4%). Similarly, on ARC-c and HellaSwag, our method maintains performance levels comparable to the high-retention baselines. This indicates that by optimizing the joint reconstruction error rather than individual expert importance, SCHUR-A* retains the model's ability to handle complex semantic dependencies, preventing the "lobotomy" effect often observed in aggressive model pruning.

Perhaps the most remarkable result is that SCHUR-A* achieves near-parity with the 25% pruning baselines while removing nearly **1.7× more parameters** (42% vs. 25%). The performance gap between our method (Avg. 66.1%) and the 25% REAP baseline (Avg. 66.9%) is negligible ($< 0.8\%$), yet we achieve substantially higher memory savings. This suggests that the additional 17% of experts retained by the baselines contribute minimally to the model's representational power. SCHUR-A* effectively identifies

this "dead weight," pushing the model towards a superior Pareto frontier where high sparsity does not come at the cost of reasoning capability.

### 4.3. Ablations and Analysis

**Inference Strategy** Upon convergence, SCHUR-A* yields an optimized expert subset $S^*$ alongside the linear refitting coefficients $\beta^*$ that minimize reconstruction error on the calibration set. This output necessitates a critical design choice for deployment, as illustrated in Figure 4: we must determine whether to explicitly apply these static coefficients to scale expert outputs (Mask-and-Scale), or to strictly rely on the topological structure $S^*$ while reverting to the original, re-normalized dynamic gating weights (Mask-Only). While $\beta^*$ guarantees the mathematical optimality of the reconstruction on observed data, a key question remains whether this static calibration transfers effectively to unseen test distributions, or if the pre-trained router's dynamic manifold provides superior generalization. To answer this, we conduct a controlled comparison on two distinct architectures: Qwen3-30B-A3B and ERNIE-4.5-21B-A3B.

Table 2 reveals a consistent trend across both architectures. Although the refitted coefficients $\beta$ are mathematically optimal for the calibration samples, they lead to overfitting. In contrast, the Mask-Only strategy significantly improves performance:

- **Perplexity Reduction:** For Qwen3-30B, using original weights reduces PPL from 8.84 to 8.25. Similarly, ERNIE-4.5-21B sees an improvement from 8.12 to 7.55.

- **Knowledge Retention:** The improvement is most pronounced in knowledge-intensive tasks. For instance, MMLU accuracy on Qwen3 jumps by +4.4%($0.579 \rightarrow$ 0.623) when switching to Mask-Only, with variance remaining stable ($\pm 0.004$).

*Table 2.* **Inference Strategy Selection.** We compare the performance (Mean ± Std) of using refitted coefficients (Mask-and-Scale) versus retaining original weights (Mask-Only). The Mask-Only strategy consistently achieves lower Perplexity (PPL) and higher accuracy with stable variance. For instance, on Qwen3, MMLU accuracy improves significantly ($0.579 \rightarrow 0.623$), indicating that original weights possess superior generalization capabilities.

| Benchmarks | PPL ↓ | ARC-c | ARC-e | BoolQ | HellaS | MMLU | OBQA | RTE | WinoG | Avg. |
|---|---|---|---|---|---|---|---|---|---|---|
| *Architecture 1: Qwen3-30B-A3B* | | | | | | | | | | |
| Unpruned Baseline | 7.74 | 0.563 | 0.790 | 0.887 | 0.778 | 0.779 | 0.454 | 0.816 | 0.702 | 0.721 |
| Mask-and-Scale | 8.84 | 0.465 ± 0.014 | 0.714 ± 0.009 | 0.854 ± 0.006 | 0.747 ± 0.004 | 0.579 ± 0.004 | 0.438 ± 0.022 | 0.780 ± 0.025 | 0.689 ± 0.013 | 0.658 ± 0.005 |
| Mask-Only | 8.25 | 0.491 ± 0.014 | 0.662 ± 0.010 | 0.866 ± 0.006 | 0.760 ± 0.004 | 0.623 ± 0.004 | 0.452 ± 0.022 | 0.740 ± 0.026 | 0.699 ± 0.013 | 0.661 ± 0.005 |
| *Architecture 2: ERNIE-4.5-21B-A3B* | | | | | | | | | | |
| Unpruned Baseline | 5.85 | 0.564 | 0.782 | 0.873 | 0.813 | 0.737 | 0.462 | 0.812 | 0.724 | 0.721 |
| Mask-and-Scale | 8.12 | 0.473 ± 0.015 | 0.671 ± 0.010 | 0.830 ± 0.007 | 0.699 ± 0.005 | 0.564 ± 0.004 | 0.398 ± 0.022 | 0.762 ± 0.026 | 0.683 ± 0.013 | 0.653 ± 0.005 |
| Mask-Only | 7.55 | 0.485 ± 0.015 | 0.682 ± 0.010 | 0.844 ± 0.006 | 0.753 ± 0.004 | 0.579 ± 0.005 | 0.454 ± 0.022 | 0.780 ± 0.023 | 0.710 ± 0.013 | 0.661 ± 0.005 |

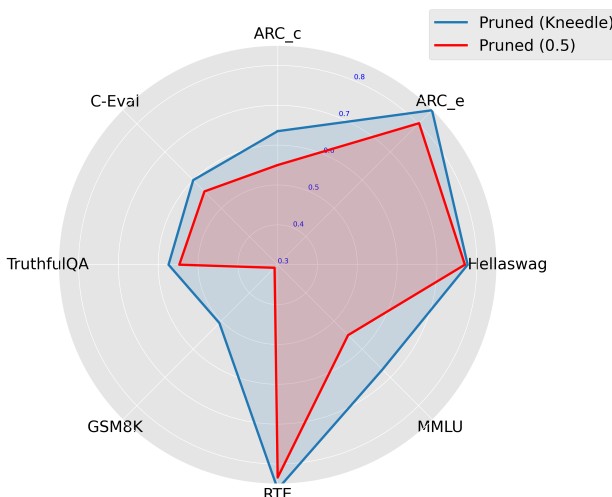

*Figure 5.* **Performance comparison between Adaptive Budgeting (Kneedle) and Fixed Pruning (50%) on Qwen3-30B-A3B-Instruct-2507.** The Kneedle algorithm (blue bars) consistently outperforms or matches the fixed ratio strategy (grey bars) across 11 standard benchmarks, particularly in complex reasoning tasks like MMLU and GSM8K.

These results confirm that while our Schur-based metric is excellent for identifying the optimal expert topology, the original pre-trained weights encode robust semantic patterns that generalize better than calibration-fitted parameters. Consequently, we adopt Mask-Only as the default configuration.

**Effectiveness of Adaptive Budgeting** To validate the necessity of our layer-wise auto-budgeting mechanism, we compare our adaptive Kneedle strategy against a fixed 50% pruning ratio baseline on the Qwen3-30B-A3B-Instruct-2507 model.

As shown in Figure 5, the adaptive strategy consistently outperforms the rigid baseline across 8 benchmarks. The advantage is particularly pronounced in knowledge-intensive and reasoning tasks: on MMLU, the adaptive method im-

proves accuracy by 12.2% ($0.550 \rightarrow 0.672$), and on GSM8K, it even leads by 19.6%. While performance on simpler tasks (e.g., RTE) remains comparable, the fixed ratio suffers catastrophic drops on complex benchmarks. Notably, the final pruning ratio for Kneedle is 41.36%, compared to 44.79% for the fixed baseline. These results confirm that redundancy distributions vary significantly across layers, making the Kneedle-based dynamic allocation essential for preserving the model's core capabilities.

**Fair Comparison under Fixed and Adaptive Budgets** The main results in Table 1 operate under budgets determined independently per method, which may conflate the quality of the pruning criterion with the advantage of an adaptive budget. To disentangle these two factors, we conduct two complementary controlled evaluations on Qwen3-30B-A3B. **(1) Fixed-Budget Setting.** We apply all baseline pruning criteria at the identical 42% pruning ratio used by SCHUR-A*, ensuring equal memory savings across all methods. **(2) Uniformly Adaptive Setting.** We apply the Kneedle algorithm to all baselines as well, giving every method the same automated layer-wise budget selection procedure. As shown in Figure 6, SCHUR-A* leads under both protocols. Under the fixed budget, even the strongest baseline (42%-REAP, Avg. 0.645) is outperformed by **1.6%**. Extending the identical Kneedle algorithm to all baselines does not close this gap (kne-REAP, Avg. 0.642), demonstrating that the performance advantage is attributable to the global combinatorial search objective rather than to a more favorable expert retention ratio.

**Global A* vs. Local Greedy** A core contribution of our framework is the deployment of the A* algorithm to perform a global search over the expert combinatorial space within each layer. To rigorously justify the computational overhead of this search, we conduct a controlled ablation study against a Greedy Baseline.

Crucially, to isolate the benefit of the search strategy itself, both methods utilize the **identical gain metric** derived

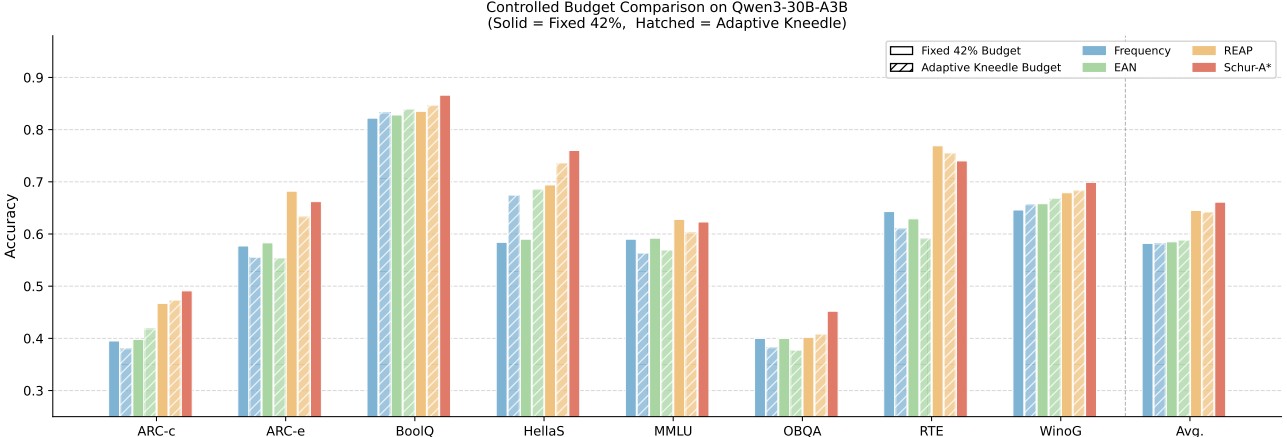

*Figure 6.* **Controlled Budget Comparison on Qwen3-30B-A3B.** Solid bars: all methods at a fixed 42% pruning budget. Hatched bars: all methods under uniformly applied Kneedle-determined adaptive budgets. SCHUR-A\* (red) consistently achieves the highest average accuracy under both protocols, confirming that its advantage stems from the quality of global combinatorial search rather than a favorable budget allocation.

from our Schur-complement formulation (specifically, the marginal gain $N^2/D$). The distinction lies solely in the optimization trajectory:

- Greedy Baseline: Iteratively selects the single expert with the highest immediate gain at each step, freezing the choice without backtracking. This represents a local optimization approach.

- SCHUR-A\*: Uses the gain metric to guide a global frontier search, allowing for backtracking to explore high-potential combinations that might be missed by myopic greedy choices.

By sharing the same objective function, this comparison directly answers the question: Is the superior performance due to the metric, or the global search?

*Table 3.* **Global Search vs. Greedy Selection.** Comparison between the Greedy baseline and SCHUR-A\*. SCHUR-A\* drastically reduces Perplexity (8.25 vs. 10.30) and achieves superior accuracy on complex tasks, particularly MMLU (+4.3%) and ARC-c (+4.2%), proving that global search is essential for preserving the model's knowledge manifold.

| Method | PPL ↓ | ARC-c | HellaSwag | MMLU | OpenBookQA |
|---|---|---|---|---|---|
| Qwen3-30B (Original) | 7.74 | 0.563 | 0.778 | 0.779 | 0.454 |
| Greedy Baseline | 10.30 | 0.449 | 0.739 | 0.580 | 0.432 |
| SCHUR-A\* (Ours) | **8.25** | **0.491** | **0.760** | **0.623** | **0.452** |
| *Improvement over Greedy* | *-2.05* | *+4.2%* | *+2.1%* | *+4.3%* | *+2.0%* |

Table 3 reveals the critical instability of the greedy approach. The Greedy baseline causes the PPL to spike to 10.30, indicating significant damage to the language probability distribution. In contrast, SCHUR-A\* maintains a PPL of 8.25, staying remarkably close to the original model (7.74). This confirms that the global search effectively minimizes the reconstruction error of the Hessian matrix within each layer, preserving the fine-grained dependencies that greedy heuristics ignore.

The advantage of global search is most pronounced in knowledge-intensive and logic-heavy tasks. While Greedy selection stagnates on MMLU (0.580) and ARC-c (0.449), SCHUR-A\* boosts performance to 0.623 (+4.3%) and 0.491 (+4.2%) respectively. This suggests that "hard" knowledge is often distributed across complex combinations of experts that individually appear less important (low greedy gain) but are collectively essential. SCHUR-A\*'s backtracking mechanism successfully retrieves these critical expert clusters, ensuring that the pruned model retains its advanced reasoning capabilities.

**Empirical Verification of Selection Optimality**  While SCHUR-A\* mathematically guarantees layer-wise optimality for the reconstruction objective, we further verify its effectiveness by comparing the selected expert subsets against a ground-truth "optimal" set obtained via brute-force enumeration on the Mixtral-8x7B architecture. Following the methodology of Lu et al. (2024), we measure the *Overlap Similarity*—the percentage of experts shared between a pruning criterion and the enumerated set.

As shown in Table 4, simple heuristics like Frequency and EAN exhibit significant mismatch with the true optimal subset, failing to capture up to 20% of essential experts. In contrast, SCHUR-A\* achieves an overlap of **92.4%** on C4 and **95.2%** on Math calibration sets. This high fidelity confirms that our Schur-complement-based search successfully recovers the combinatorial synergies discovered by exhaustive search, but at a fraction of the computational cost ($\mathcal{O}(|S|^2 m)$ vs. $\mathcal{O}(\binom{E}{r})$).

*Table 4.* **Similarity with Brute-Force Enumerated Optimal Sets (Mixtral-8x7B).** We compare the expert selection overlap between various criteria and the exact optimum identified via enumeration (Lu et al., 2024). SCHUR-A* achieves near-perfect alignment with the global optimum.

| Calibration Set | Frequency | EAN | Schur-A* |
|---|---|---|---|
| C4 | 79.20% | 86.10% | **92.40%** |
| Math | 82.40% | 88.50% | **95.20%** |

### 4.4. Efficiency and Sensitivity Analysis

We conclude our experimental evaluation by analyzing the computational efficiency of the search process and the structural sensitivity of specific MoE layers.

In terms of computational efficiency, SCHUR-A* proves to be highly scalable. We clarify that the search across all layers is executed in parallel, meaning the total wall-clock time is governed by the single most time-consuming layer. While a granular analysis reveals distinct behaviors contingent on layer depth, for the vast majority of MoE layers, SCHUR-A* typically converges to the globally optimal subset within 1–2 minutes on a single NVIDIA A100 GPU, with even the most time-consuming layers completing within 10 minutes. More importantly, we note that the dominant time cost in expert pruning—for both SCHUR-A* and any greedy baseline—is not the search process itself, but the shared prerequisite: forward inference over the calibration corpus to collect activation statistics and Gram matrices. Once the calibration data is prepared, the search overhead is marginal by comparison. However, special considerations remain for specific stages: in the initial layers (Layers 0–2), the dense feature entanglement of early network stages requires careful handling, while the final MoE layer presents a challenge of numerical precision due to highly specialized activation patterns that can induce ill-conditioned Gram matrices, occasionally destabilizing the spectral bounds.

To ensure stability without sacrificing the overall compression rate, we adopt a hybrid configuration: we restore the first three layers and the final layer to their full capacity ($N = 128$), while applying the optimized pruning results to all intermediate layers. Under this configuration, SCHUR-A* achieves a final expert retention rate of 57.89% on Qwen3-30B-A3B. Despite the prolonged search in early layers, the total wall-clock time remains practical for offline compression, further accelerated by our vectorized gain updates and parallel execution across layers.

## 5. Limitations

Despite the empirical effectiveness and theoretical grounding of SCHUR-A*, we acknowledge several limitations inherent to our current framework that merit further investiga-

tion.

**Numerical Stability in Ill-Conditioned Regimes.** In half-precision regimes, correlated experts can yield ill-conditioned Gram matrices, risking Cholesky decomposition failures. Although we employ ridge regularization ($\epsilon$-damping) and selective capacity retention as mitigations, adopting robust formulations like Singular Value Decomposition (SVD) remains a desirable direction for guaranteed numerical stability without manual intervention.

**Physical Interpretation of Unconstrained Refitting.** Unconstrained refitting yields coefficients $\beta$ that may violate the probabilistic constraints of MoE gating, explaining the superior generalization of *Mask-Only* inference. Future integration of Non-Negative Least Squares (NNLS) constraints could resolve this discrepancy between the optimization objective and the physical interpretation of routing weights.

**Layer-wise vs. End-to-End Optimality.** We guarantee optimality strictly for the local reconstruction objective, not end-to-end task performance. This greedy layer-wise approach implicitly ignores inter-layer dependencies, assuming local error minimization approximates global loss reduction—a widely-accepted surrogate in established post-training compression frameworks such as GPTQ (Frantar et al., 2022), SparseGPT (Frantar & Alistarh, 2023), and Wanda (Sun et al., 2023). Future research might explore iterative block-wise optimization to address these global relaxation effects.

## 6. Conclusion

In this paper, we presented SCHUR-A*, a rigorous framework that reformulates post-training MoE pruning as a combinatorial optimization problem solved via a layer-wise optimal A* search. By leveraging an exact Schur-complement-based gain metric implemented through a high-performance Vectorized Update Kernel, we successfully reduced the computational complexity from cubic to quadratic, making global search feasible for large-scale models. Integrated with adaptive Kneedle budgeting, our approach bridges the gap between theoretical optimality and practical efficiency. Our findings confirm that preserving high-order expert correlations is paramount for maintaining model fidelity, offering a robust pathway for deploying massive sparse models on resource-constrained devices without the need for retraining.

## Acknowledgments

This work was supported by the Sichuan Provincial Natural Science Foundation Project(No. 2025ZNSFSC0508).

## Impact Statement

This paper presents SCHUR-A\*, a post-training expert pruning method for sparse Mixture-of-Experts (MoE) language models that significantly reduces inference-time memory consumption while preserving model capability. By mitigating the memory wall that often restricts the deployment of large MoE models, this work has the potential to broaden access to high-capacity language models beyond environments with large-scale computational infrastructure.

Reducing memory requirements may enable large expert-based models, including those at the tens-of-billions parameter scale, to be deployed on more affordable hardware. This can lower barriers for academic researchers, small organizations, and developers with limited resources, contributing to a more inclusive and diversified ecosystem for large language model research and applications. In addition, because SCHUR-A\* operates entirely in a post-training setting and does not require retraining or extensive fine-tuning, it offers a low-cost approach to model compression that can reduce energy consumption and support more sustainable deployment of large models.

At the same time, pruning decisions in SCHUR-A\* rely on statistics derived from a calibration dataset. If this dataset does not adequately represent certain domains, languages, or user populations, the pruned model may underperform in these settings, potentially discarding experts that capture long-tail or minority-specific behaviors. This highlights the importance of careful calibration data selection and thorough evaluation prior to deployment, particularly in high-stakes or safety-critical applications. We emphasize that SCHUR-A\* is intended as a systems optimization tool, and responsible use requires transparency, appropriate oversight, and consideration of downstream impacts.

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

# A. Appendix

### A.1. Optimality and Tightness Analysis

We provide a rigorous theoretical guarantee for the proposed heuristic and empirically validate its search efficiency using the Qwen3-MoE model family.

**Layer-wise Optimality Guarantee.** The convergence of A* search to a layer-wise globally optimal solution necessitates an *admissible* heuristic. As derived in Sec. 3.2, our function $h(S, t)$ serves as a principled upper bound on the incremental reconstruction reward. Consequently, SCHUR-A* satisfies the admissibility criterion, mathematically ensuring that the first complete expert subset identified is the layer-wise global optimum.

**Empirical Tightness and Search Efficiency.** The practical efficiency of A* is governed by the *tightness* of the bound; a near-exact bound facilitates aggressive pruning of suboptimal branches. To quantify this, we define the **Consistency Ratio** $\rho$:

$$\rho = \frac{F(S^*)}{h(S_{root})} \tag{15}$$

where $F(S^*)$ is the true optimal reward and $h(S_{root})$ is the initial spectral bound at the root node. A ratio $\rho \to 1$ indicates a near-perfect estimation of the potential gain.

Figure 7 illustrates the distribution of $\rho$ across all layers for both **Qwen3-30B-A3B** and its **Instruct** variant. The empirical results reveal exceptional tightness: the base model yields a median ratio of **0.9857**, while the Instruct model achieves **0.9816**. This minimal overestimation (less than 2%) confirms that the Schur-complement relaxation effectively captures the latent expert correlations. This high-fidelity guidance allows SCHUR-A* to collapse the search space significantly, making exact subset selection computationally feasible for large-scale MoEs.

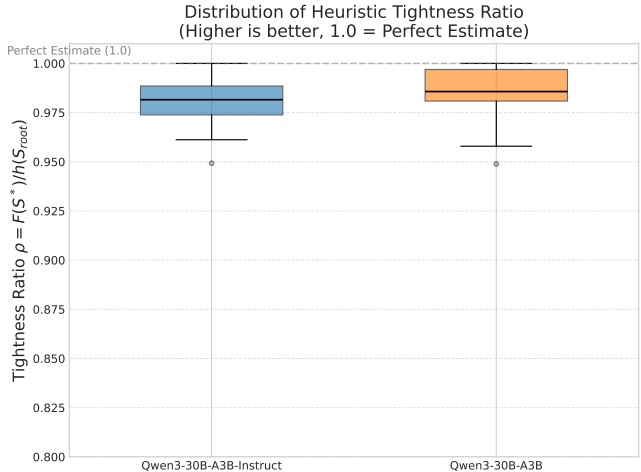

*Figure 7.* **Empirical Analysis of Heuristic Tightness.** The box plot compares the Consistency Ratio $\rho$ across layers. Median ratios near 1.0 (Base: 0.9857, Instruct: 0.9816) indicate that the Schur-complement heuristic provides a near-exact upper bound, ensuring near-optimal pruning efficiency.

### A.2. Generalization to Other Architectures

To further verify the robustness and versatility of SCHUR-A*, we extend our evaluation to two additional architectures with distinct expert granularities and routing mechanisms: **ERNIE-4.5-21B-A3B** and **DeepSeek-V2-Lite**. These models represent diverse MoE paradigms outside the Qwen family. Notably, the Kneedle algorithm automatically adapts to the varying redundancy profiles of these architectures, determining conservative pruning ratios of 32.93% for ERNIE and approximately 39% for DeepSeek-V2-Lite. All methods are evaluated under a uniformly applied adaptive budget protocol to ensure a fair, "apples-to-apples" comparison across diverse topologies.

*Table 5.* **Performance Comparison on ERNIE-4.5-21B-A3B.** We compare SCHUR-A* against standard baselines. **Note:** Our method operates at an adaptive pruning ratio of **32.93%**. It maintains high fidelity across all tasks, surpassing 50% baselines and matching the performance of 25% baselines.

| Method | ARC-c | ARC-e | BoolQ | HellaS | MMLU | OBQA | RTE | WinoG | Avg. |
|---|---|---|---|---|---|---|---|---|---|
| ERNIE-4.5-21B (Original) | 0.564 | 0.782 | 0.873 | 0.813 | 0.737 | 0.462 | 0.812 | 0.724 | 0.721 |
| *Baselines at 25% Pruning Ratio* | | | | | | | | | |
| Frequency | $0.486 \pm 0.004$ | $0.711 \pm 0.000$ | $0.852 \pm 0.004$ | $0.675 \pm 0.003$ | $0.628 \pm 0.003$ | $0.373 \pm 0.003$ | $0.780 \pm 0.006$ | $0.676 \pm 0.005$ | $0.648 \pm 0.001$ |
| EAN | $0.498 \pm 0.005$ | $0.713 \pm 0.002$ | $0.863 \pm 0.002$ | $0.717 \pm 0.004$ | $0.625 \pm 0.001$ | $0.405 \pm 0.011$ | $0.811 \pm 0.009$ | $0.702 \pm 0.005$ | $0.667 \pm 0.000$ |
| REAP | $0.527 \pm 0.004$ | $0.759 \pm 0.002$ | $0.857 \pm 0.003$ | $0.717 \pm 0.003$ | $0.644 \pm 0.001$ | $0.409 \pm 0.009$ | $0.756 \pm 0.008$ | $0.690 \pm 0.001$ | $0.670 \pm 0.002$ |
| *Baselines at 50% Pruning Ratio* | | | | | | | | | |
| Frequency | $0.400 \pm 0.002$ | $0.584 \pm 0.006$ | $0.830 \pm 0.001$ | $0.522 \pm 0.003$ | $0.506 \pm 0.006$ | $0.303 \pm 0.004$ | $0.758 \pm 0.004$ | $0.625 \pm 0.004$ | $0.566 \pm 0.002$ |
| EAN | $0.417 \pm 0.005$ | $0.633 \pm 0.005$ | $0.830 \pm 0.003$ | $0.572 \pm 0.003$ | $0.509 \pm 0.002$ | $0.336 \pm 0.003$ | $0.785 \pm 0.014$ | $0.626 \pm 0.003$ | $0.589 \pm 0.003$ |
| REAP | $0.417 \pm 0.009$ | $0.626 \pm 0.007$ | $0.803 \pm 0.006$ | $0.556 \pm 0.003$ | $0.505 \pm 0.003$ | $0.325 \pm 0.006$ | $0.775 \pm 0.014$ | $0.623 \pm 0.008$ | $0.579 \pm 0.002$ |
| **SCHUR-A* (Ours)** | $0.485 \pm 0.015$ | $0.682 \pm 0.010$ | $0.844 \pm 0.006$ | $0.753 \pm 0.004$ | $0.579 \pm 0.005$ | $0.454 \pm 0.022$ | $0.780 \pm 0.023$ | $0.710 \pm 0.013$ | $\mathbf{0.661} \pm 0.005$ |

**ERNIE-4.5-21B-A3B.** Table 5 presents the performance comparison on the ERNIE model. Consistent with our main findings, SCHUR-A* achieves a superior trade-off. With an average accuracy of **66.1%**, our method significantly outperforms all 50% baselines (which max out at 58.9%) and remains highly competitive with the 25% baselines (e.g., Frequency at 64.8%), effectively bridging the gap between high-sparsity and high-fidelity regimes.

*Table 6.* **Generalization to DeepSeek-V2-Lite (Adaptive Kneedle Budget).** SCHUR-A* consistently outperforms all ranking-based baselines. While absolute accuracy drops are larger than on Qwen3/ERNIE, this reflects the intrinsically higher pruning sensitivity of DeepSeek-V2-Lite rather than a failure of the search strategy.

| Method | ARC-c | ARC-e | BoolQ | HellaS | MMLU | OBQA | RTE | WinoG | Avg. |
|---|---|---|---|---|---|---|---|---|---|
| DeepSeek-V2-Lite (Unpruned) | 0.464 | 0.742 | 0.801 | 0.778 | 0.554 | 0.446 | 0.606 | 0.708 | 0.637 |
| kne-Frequency | 0.258 | 0.353 | 0.503 | 0.455 | 0.240 | 0.280 | 0.545 | 0.493 | 0.391 |
| kne-EAN | 0.300 | 0.470 | 0.527 | 0.568 | 0.251 | 0.320 | 0.473 | 0.517 | 0.428 |
| kne-REAP | 0.308 | 0.491 | 0.513 | 0.584 | 0.279 | 0.317 | 0.479 | 0.533 | 0.438 |
| **SCHUR-A* (Ours)** | **0.304** | **0.516** | **0.514** | **0.588** | **0.264** | **0.358** | **0.487** | **0.552** | **0.448** |

**DeepSeek-V2-Lite.** Table 6 displays the results for the DeepSeek architecture. SCHUR-A* achieves the highest average accuracy (0.448) among all pruned methods. The larger absolute drops relative to Qwen3/ERNIE are consistent with DeepSeek-V2-Lite exhibiting higher sensitivity to expert removal at this compression level.

## A.3. Calibration Sensitivity Analysis

To assess the robustness of SCHUR-A* to calibration data, we perform sensitivity analyses across two dimensions: dataset size and domain composition.

**Calibration Set Size.** The choice of 512 sequences from C4 was intended as a conservative setting. To verify this, we compare performance across different calibration set sizes (128 vs. 256 vs. 512 sequences) on Qwen3-30B-A3B. As shown in Table 7, SCHUR-A* is remarkably robust across these settings, with 128 sequences (a common standard in LLM pruning) already providing stable results. This indicates that larger calibration sets are not strictly necessary for achieving optimal selection.

*Table 7.* **Calibration Size Sensitivity on Qwen3-30B-A3B.** Performance remains stable across different sequence counts, suggesting that 512 sequences is a conservative margin rather than a heavily tuned optimum.

| Seq Count | ARC-c | ARC-e | BoolQ | HellaS | MMLU | OBQA | RTE | WinoG | Avg. |
|---|---|---|---|---|---|---|---|---|---|
| 128 | 0.491 | 0.717 | 0.869 | 0.758 | 0.627 | 0.434 | 0.801 | 0.683 | 0.673 |
| 256 | 0.490 | 0.687 | 0.865 | 0.771 | 0.628 | 0.429 | 0.782 | 0.678 | 0.666 |
| 512 | 0.491 | 0.662 | 0.866 | 0.760 | 0.623 | 0.452 | 0.740 | 0.699 | 0.661 |

**Calibration Domain Composition.**    While general knowledge is preserved using C4, specialized reasoning tasks (e.g., Code, Math) may exhibit higher sensitivity to the calibration distribution. We compare our default C4 calibration against a domain-specific set comprising a 1:1 mix of Code and GSM8K data.

*Table 8.* **Domain Sensitivity on Specialized Tasks (Qwen3-30B-A3B, $\approx$42% pruning).** Proper domain-matched calibration significantly recovers performance on reasoning-intensive benchmarks, though extreme stress tests like AIME24 reveal intrinsic structural costs of aggressive pruning.

| Methodology | HumanEval | MBPP | MATH-500 | GSM8K | AIME24 |
|---|---|---|---|---|---|
| Unpruned Baseline | 34.3% | 72.2% | 56.2% | 89.3% | 13.3% |
| SCHUR-A* (C4 Calib.) | $\approx$0.0% | $\approx$0.0% | $\approx$0.0% | $\approx$50.7% | $\approx$0.0% |
| **SCHUR-A* (Code/GSM8K Calib.)** | **32.3%** | **50.1%** | **48.0%** | **78.9%** | **3.3%** |

As illustrated in Table 8, domain-specific calibration enables a substantial recovering of performance, rescuing the model from catastrophic failure on code and math tasks. Specifically, the domain-adapted model retains nearly 90% of baseline GSM8K accuracy and over 94% of HumanEval performance. However, on frontier tasks like AIME24, a gap remains even with domain matching, suggesting that aggressive expert pruning at 42% ratio inevitably disrupts some intricate reasoning chains that calibration alone cannot resolve.

### A.4. Efficiency and Retention Profiles

In this section, we analyze the computational efficiency of the SCHUR-A* search process. As illustrated in Figure 8, the proposed vectorized gain computation significantly accelerates the node expansion phase by processing all candidate experts simultaneously. In a naive greedy search, the linear system must be re-solved for each of the $m$ candidates individually, leading to a prohibitive total complexity of $\mathcal{O}(m \cdot |S|^3)$. In contrast, by leveraging cached Cholesky factors $L_S$, our vectorized implementation evaluates all candidates in three efficient stages: (1) computing projection coefficients $W \in \mathbb{R}^{|S| \times m}$ via a single triangular solve; (2) deriving the Schur complement denominators from the column-wise norms of $W$; and (3) calculating exact gains through parallel operations. This strategy reduces the dominant complexity per step to $\mathcal{O}(m \cdot |S|^2)$, providing a speedup factor of roughly $|S| \times$ and allowing the search to scale to fine-grained architectures with over 128 experts.

Furthermore, we examine the expert retention profiles identifying how redundancy is distributed across the network hierarchy. Figure 9 plots the number of retained experts ($r_l$) for Qwen3-30B and ERNIE-4.5-21B. The distribution reveals a distinct "U-shaped" pattern, primarily because we manually roll back the initial layers (0–2) and the final layer to full capacity. This restoration is a necessary fallback as these layers frequently encounter numerical singularity during the subset selection process, making the reconstruction objective ill-conditioned. Our analysis thus focuses on the intermediate layers, where the Kneedle algorithm identifies significant redundancy. Notably, when accounting for the total expert count ($E = 128$ for Qwen3 vs. $E = 64$ for ERNIE), Qwen3-30B exhibits a lower retention ratio (avg. $\sim$57%) compared to ERNIE-4.5-21B (avg. $\sim$63%), identifying a higher degree of proportional redundancy in its fine-grained expert pool. This confirms that intermediate layers across different architectures contain substantial "dead weight" that can be automatically minimized while preserving the model's core representational power.

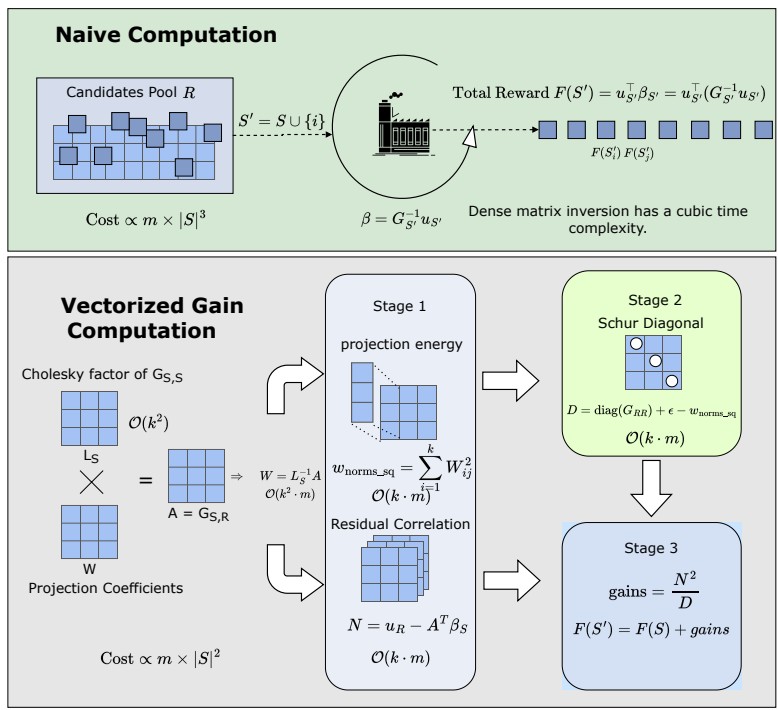

*Figure 8.* **Computational Efficiency Comparison: Naive vs. Vectorized Search.** We contrast the sequential bottleneck of standard greedy approaches with our efficient vectorized pipeline that leverages cached Cholesky factors for parallel gain calculation.

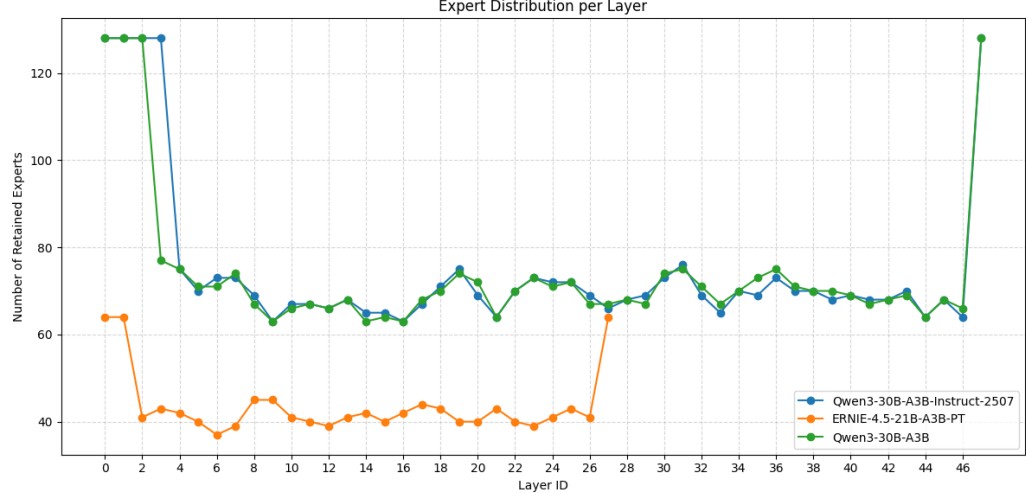

*Figure 9.* **Layer-wise Expert Retention Profiles via Adaptive Budgeting.** We illustrate the number of retained experts across all layers, highlighting the U-shaped distribution and the model-specific compressibility identified by the Kneedle algorithm.

