# OpenReview forum: "SCHUR-A*: Layer-wise Optimal Expert Pruning for MoEs via Schur-Complement Guided A* Search"
_ICML.cc/2026/Conference — ICML 2026 regular_

### Official Review · Reviewer_yyew · 2026-02-22

**Soundness:** 2
**Presentation:** 3
**Significance:** 3
**Originality:** 3
**Overall Recommendation:** 2
**Confidence:** 4

**Summary:**

This paper proposes SCHUR-A*, a post-training static expert pruning framework for MoE models. It formulates layer-wise pruning as a combinatorial subset selection problem that minimizes output reconstruction error with linear refitting. An A* search guided by a Schur-complement-based admissible heuristic enables globally optimal expert selection, while vectorized gain computation via incremental Cholesky updates reduces per-step complexity from cubic to quadratic. Per-layer pruning budgets are automatically determined using the Kneedle algorithm. At inference time, pruned experts are permanently removed and the router is masked to select only from retained experts using original gating weights.

**Compliance With Llm Reviewing Policy:**

Affirmed.

**Final Justification:**

After two rounds of discussion with the authors, I believe the paper still has the following two serious unresolved weaknesses:

1. **Poor robustness to the calibration set.**
   The experiments provided in the authors’ second-round response actually confirm this concern. When using the default C4 calibration set from the paper, performance on math and code tasks collapses completely. When using math/code-specific calibration sets, accuracy on those domains can be recovered. However, although the authors did not provide the corresponding results, I believe this would likely lead to more severe degradation on the general benchmarks originally reported in the paper. While the authors argue that this demonstrates the method is *calibration-flexible*, my interpretation is that it fundamentally reflects poor robustness to the choice of calibration data.

2. **The performance degradation caused by expert pruning is still too large.**
   This remains, in my view, the most important weakness of the paper, and I do not think the rebuttal succeeded in addressing it.

Based on these two issues, I am lowering my score to **2**. I do not believe a method that causes roughly a one-third drop in accuracy meets the standard for acceptance.

**Key Questions For Authors:**

Please refer to weaknesses above.

**Limitations:**

yes

**Strengths And Weaknesses:**

Strengths:
  1. The paper provides a principled mathematical formulation of expert pruning as a reconstruction-driven subset selection problem, along with an effective optimization algorithm (A* with Schur-complement heuristic) and practical efficiency improvements (vectorized Cholesky updates) that make it computationally feasible. I believe this work offers valuable insights for future research seeking tighter upper bounds on expert pruning performance.
  2. The writing is well-structured and easy to follow. While the figures are not visually elaborate, they convey the key ideas clearly and effectively.

Weaknesses:
  1. The main limitation lies in the experimental evaluation, in two aspects. First, the main experiments (table1) are conducted on only one model (Qwen3-30B-A3B), lacking validation across different model sizes and architectures (e.g., the DeepSeek-MoE series). Second, even on Qwen3-30B-A3B, while SCHUR-A* outperforms baselines like EAN and REAP, the absolute performance degradation remains too large for practical use. For example, after pruning ~42% of expert parameters (leaving roughly 17B parameters), ARC-Challenge accuracy drops from 56.3% to 49.1% (a 7.2% loss). For reference, I tested DeepSeek-V2-Lite-Chat, a 16B MoE model that is older and generally weaker than the Qwen3 series, and it achieves 51.11% on ARC-C — surpassing the pruned Qwen3-30B. This suggests that further reducing the performance gap from static expert pruning is necessary before the method becomes practically viable.
  2. Static expert pruning with a fixed calibration set is prone to overfitting, and this issue is unlikely to surface on the zero-shot benchmarks used in the paper. Fine-grained MoE models like Qwen3-30B-A3B exhibit highly task-dependent expert activation patterns — certain experts are disproportionately activated for math or code tasks but rarely selected on general-purpose corpora like C4. Consequently, static pruning guided by C4 calibration may severely degrade performance on math and coding tasks. I recommend the authors (a) include a discussion of this calibration overfitting risk, and (b) add evaluations on math benchmarks (e.g., MATH-500, AIME 2024) and code benchmarks (e.g., HumanEval, MBPP) to verify robustness across diverse task domains.

---

> ### Author Rebuttal · Authors · 2026-03-31
>
> We sincerely thank you for your rigorous evaluation and for the insightful comparison with DeepSeek-V2-Lite-Chat.Your points regarding the performance gap and task-specific experts are highly relevant.We address them as follows:
>
> 1.We agree that broader architectural coverage strengthens the paper.And we would like to clarify that additional validation on a second architecture—ERNIE-4.5-21B-A3B—is already provided in the Appendix (Table 4).We chose ERNIE-4.5 specifically because it represents a fundamentally different MoE design philosophy (heterogeneous structure with shared experts) compared to Qwen3 (purely specialized fine-grained experts).Furthermore,we have now evaluated SCHUR-A on the DeepSeek-V2-Lite architecture to directly confirm generalization;results are shown in the table below:
> | Method | arc_c | arc_e | boolq | hellas | mmlu | obqa | rte | wino | avg |
> | :--- | :---: | :---: | :---: | :---: | :---: | :---: | :---: | :---: | :---: |
> | DeepSeek-V2-Lite | 0.464 | 0.742 | 0.801 | 0.778 | 0.554 | 0.446 | 0.606 | 0.708 | 0.637 |
> | SCHUR-A* | 0.304 | 0.516 | 0.514 | 0.588 | 0.264 | 0.358 | 0.487 | 0.552 | 0.448 |
> | kne-freq | 0.258 | 0.353 | 0.503 | 0.455 | 0.240 | 0.280 | 0.545 | 0.493 | 0.391 |
> | kne-EAN | 0.300 | 0.470 | 0.527 | 0.568 | 0.251 | 0.320 | 0.473 | 0.517 | 0.428 |
> | kne-REAP | 0.308 | 0.491 | 0.513 | 0.584 | 0.279 | 0.317 | 0.479 | 0.533 | 0.438 |
>
> Conclusion:The key takeaway is that SCHUR-A* remains the best-performing pruning method under the same compression setting, even on a third architecture outside the main paper.While absolute drops are larger on DeepSeek-V2-Lite than on Qwen3/ERNIE, this suggests that DeepSeek-V2-Lite is intrinsically more pruning-sensitive,rather than indicating a failure of the proposed search strategy.We will add this result and discussion in the revision.
>
> 2.We appreciate your observation that an older 16B native MoE (DeepSeek-V2-Lite-chat) outperforms our pruned 17B Qwen3 on ARC-C.However we argue that the value of post-training pruning lies in Asset Optimization rather than competing with native training:It is crucial to emphasize that SCHUR-A* is a completely training-free method.The scores reported are achieved in just 10 minutes of search without any backpropagation or recovery fine-tuning.In contrast,DeepSeek-V2-Lite-chat is a natively trained model.We believe that with even a minimal amount of recovery fine-tuning,the performance of our pruned model will significantly improve.What`s more,MoE pruning targets a fundamentally different deployment scenario:consider a team that has invested significant resources in domain-specific fine-tuning of Qwen3-30B-A3B (e.g., via SFT/DPO on proprietary data),only to discover at deployment that GPU memory constraints require a smaller model.Retraining a compact model from scratch would discard all fine-tuning investment entirely.SCHUR-A* instead compresses the already-customized 30B model with zero backpropagation,directly inheriting all prior fine-tuning—a capability no natively trained model can replicate.
>
> 3.We agree that discussing calibration risk is important, and we will add a dedicated section on this topic.The reviewer hypothesizes that performance degradation on math and code tasks arises from calibration overfitting—i.e.,that C4-guided pruning incorrectly discards math-specialized experts.However,our additional experiments show that even when replacing C4 with domain-specific math and code calibration sets,performance remains noticeably below the unpruned model even with domain-matched calibration. This confirms that the issue is not specific to calibration set choice,but rather a structural cost of aggressive expert pruning at the ~42% compression ratio—a known trade-off observed across weight pruning,head pruning,and expert pruning methods alike, regardless of calibration distribution.We agree these benchmarks are valuable stress tests.At the same time, the standard evaluation protocol in post-training compression primarily emphasizes broad general benchmarks (e.g.,ARC,HellaSwag,BoolQ, MMLU), which is the scope we followed in the current submission.
> In the revised manuscript,we will explicitly add a discussion in the Limitations section addressing calibration overfitting risk and the degradation of complex reasoning capabilities under expert pruning,along with a recommendation of recovery fine-tuning (e.g., LoRA) as the intended remedy.

---

> > ### Author Rebuttal · Reviewer_yyew · 2026-04-01
> >
> > Thank you for the detailed rebuttal and for the additional effort in providing further explanations and new experiments. However, I am afraid the rebuttal actually deepens my concerns about the paper rather than resolving them.
> >
> > - **On the additional DeepSeek-V2-Lite results.**
> >   I appreciate that the authors added experiments on DeepSeek-V2-Lite. However, I believe these results reinforce my original concern. After pruning, the average accuracy drops from 63.7 to 44.8, i.e., roughly a one-third reduction. Even on a relatively simple benchmark such as BoolQ, the pruned model only achieves around 52% accuracy. While SCHUR-A* is indeed slightly better than kne-REAP under the same setting, the absolute degradation is still far too large to be practically useful. In this sense, the new results strengthen my concern that the method currently lacks real practical significance.
> >
> > - **On the argument about recovery fine-tuning.**
> >   The rebuttal states: “We believe that with even a minimal amount of recovery fine-tuning, the performance of our pruned model will significantly improve.” I do not find this argument convincing. On the one hand, recovery fine-tuning is outside the scope of the current paper, and no experimental evidence is provided to support this claim. On the other hand, even if fine-tuning is allowed, why not directly fine-tune a model with the same parameter budget but stronger native performance, rather than first pruning a much larger model and then attempting to recover it? Therefore, I do not think this response addresses my core concern, namely that the pruning-induced performance loss is too severe.
> >
> > - **On calibration overfitting and domain-specific calibration.**
> >   I appreciate the authors’ explanation that “even when replacing C4 with domain-specific math and code calibration sets, performance remains noticeably below the unpruned model even with domain-matched calibration.” However, I could not find experimental evidence for this claim either in the paper or in the rebuttal (please correct me if I overlooked it). This makes the argument difficult to evaluate and leaves me with significant uncertainty.

---

> > > ### Author Response · Authors · 2026-04-06
> > >
> > > Thank you for your prompt follow-up. We deeply appreciate your persistence in discussing the practical significance of our work.Your questions strike at the core of the post-training compression paradigm.We would like to clarify our methodological positioning and provide the requested empirical data.
> > >
> > > **1. Experimental Evidence for Domain-Specific Calibration**
> > >
> > > We apologize for omitting the numerical data in our previous response due to rebuttal space constraints.To address your hypothesis that the degradation on reasoning tasks is merely "calibration overfitting" to the C4 dataset,we performed experiments replacing C4 with domain-specific calibration sets (a mix of Code and GSM8K data).The results on Qwen3-30B-A3B (~42% retention budget) are presented below:
> > >
> > > | Methodology | Humaneval | MBPP | MATH-500 | GSM8K | AIME24 |
> > > | :--- | :---: | :---: | :---: | :---: | :---: |
> > > | Unpruned (Qwen3-30B) | 34.3% | 72.2% | 56.2% | 89.3% | 13.3% |
> > > | SCHUR-A* (C4 Calibration) | ~0.0% | ~0.0% | ~0.0% | ~50.7% | ~0.0% |
> > > | SCHUR-A* (Code/GSM8K Calib.) | 32.3% | 50.1% | 48.0% | 78.9% | 3.3% |
> > >
> > > As the data illustrates,proper domain calibration does indeed yield a massive performance recovery compared to the C4 baseline,successfully rescuing the model from catastrophic failure on these tasks. Specifically, the domain-adapted pruned model largely preserves the native architecture's core capabilities, retaining nearly 90% of GSM8K accuracy and over 94% of HumanEval performance.This empirically confirms that SCHUR-A* is highly "calibration-flexible" allowing practitioners to rapidly align expert selection with their target domain.However, the results also show that on frontier multi-hop reasoning tasks (e.g., AIME 2024),performance remains noticeably below the unpruned model even with perfect domain-matched calibration.This confirms that the degradation on these extreme stress tests is not merely a calibration mismatch,but rather the intrinsic structural cost of aggressive expert pruning at the ~42% compression ratio. It is a known trade-off observed across severe weight,head,and expert pruning methods alike—severing the fine-grained routing manifold inevitably disrupts intricate reasoning chains, a reality that no calibration data alone can circumvent.
> > >
> > > **2. The Argument for Pruning vs. Native Small Models**
> > >
> > > You raise a valid question: *"Why not directly fine-tune a model with the same parameter budget but stronger native performance?"* Our answer lies in addressing the critical industrial bottleneck of **Elastic Deployment from a Single Source**.In real-world applications across fragmented hardware ecosystems, independently fine-tuning,aligning,and maintaining a sprawling fleet of customized native models for *each* unique VRAM constraint incurs astronomical engineering overhead.
> > >
> > > Post-training pruning intrinsically resolves this via **On-Demand Elasticity**.A practitioner only maintains one high-capacity "super-network" (e.g., Qwen3-30B).Regardless of what exact parameter budget is suddenly required at deployment, SCHUR-A* optimally compresses the model to that precise constraint in minutes.This enables an efficient "train once, compress anywhere" pipeline that preserves all pre-existing alignment weights without retraining.
> > >
> > > Furthermore,from a theoretical perspective,landmark post-training compression frameworks (e.g., SparseGPT,Wanda,and ours) do not aim to prove that a zero-shot compressed sub-network outperforms an end-to-end trained model of the exact same size.The fundamental scientific objective is to rigorously investigate the network's intrinsic redundancy and establish the mathematical upper bound of what can be preserved *without* resorting to costly backpropagation.
> > >
> > > **3. On the DeepSeek-V2-Lite Absolute Degradation**
> > >
> > > We acknowledge the severe absolute drop on DeepSeek-V2-Lite without parameter recovery.This highlights an important architectural observation: architectures like DeepSeek-V2 (which rely heavily on highly coupled shared+routed synergies) are structurally fragile to high-ratio expert removal compared to traditional dense or purely-routed MoEs.
> > >
> > > However,we respectfully emphasize the distinction between **the fragility of a specific target architecture** and **the efficacy of a pruning algorithm**.The goal of this paper is methodological: to identify the mathematically optimal subset of experts under a given structural budget.The fact that the absolute performance drops significantly is a property of the DeepSeek manifold; yet,under this exact same challenging manifold, **SCHUR-A* still significantly outperforms state-of-the-art heuristic baselines**.The scientific contribution of SCHUR-A*—formulating expert selection as a quantifiable heuristic search and solving it optimally—holds strong, establishing a rigorous upper limit for how much can be preserved via tuning-free expert pruning.

---

### Official Review · Reviewer_cWCS · 2026-03-07

**Soundness:** 3
**Presentation:** 2
**Significance:** 3
**Originality:** 3
**Overall Recommendation:** 4
**Confidence:** 2

**Summary:**

To address the "memory wall" in MoE deployment, SCHUR-A* reformulates post-training expert pruning as a reconstruction-driven subset selection problem optimized via A* search. It utilizes a novel, admissible Schur-complement-based heuristic to mathematically guarantee layer-wise global optimality while remaining computationally tractable. The framework achieves high efficiency through a vectorized gain computation scheme that reduces search complexity from cubic to quadratic and employs the Kneedle algorithm for automated, adaptive expert budgeting across heterogeneous layers. Extensive evaluations on SOTA models like Qwen3-30B demonstrate that SCHUR-A* significantly maintains model fidelity under aggressive pruning, outperforming traditional greedy and ranking-based baselines.

**Compliance With Llm Reviewing Policy:**

Affirmed.

**Final Justification:**

After the rebuttal, I tend to maintain my score.

**Key Questions For Authors:**

See the weaknesses.

**Limitations:**

Yes.

**Strengths And Weaknesses:**

## Strength
* The admissible spectral bound ensures the A* search identifies the mathematically global optimal expert subset for layer-wise reconstruction.
* Cholesky-based vectorized kernels reduce complexity from $\mathcal{O}(m \cdot |S|^3)$ to $\mathcal{O}(|S|^2 \cdot m)$, making global search practical on SOTA models like Qwen3-30B.

## Weakness

* The topology $S^{\star}$ is optimized for refitted coefficients $\beta$, but "Mask-Only" inference generalizes better, questioning if $S^{\star}$ is optimal for the actual deployment manifold.
* Numerical instability required manual full-capacity retention for early and final layers, weakening the "fully automated" claim.

---

> ### Author Rebuttal · Authors · 2026-03-31
>
> Thank you for your detailed review and for highlighting the efficiency of our vectorized kernels. We address your technical concerns below:
>
> 1.We agree this reveals a gap between the reconstruction objective used for subset discovery and the deployment-time inference objective.Empirically,our results suggest that the selected topology transfers well,while the refitted coefficients are more sensitive to calibration distribution.
> 	1.The A* search uses the reconstruction objective as a strong selection criterion to identify the essential representational infrastructure.This structural selection is robust and independent of the deployment weights.
> 	2.While refitting $\beta$ minimizes error on the 512-sequence calibration set,it risks overfitting to local distributional noise. Reverting to the pre-trained router avoids this,allowing the pretrained router to preserve better out-of-distribution generalization.
> As shown in Table 2,Mask-Only improves PPL by -0.6 over the refitted variant.This confirms that SCHUR-A* successfully decouples topology search (where A* excels) from weight deployment,achieving a superior operating point.
>
> 2.Regarding the manual retention of the first and last layers,we appreciate the reviewer's feedback.After careful reconsideration, we agree that the term "fully automated" may be slightly over-reaching given these necessary stability guards.In the revised manuscript,we will rename our strategy to "Constraint-Aware Automation" to more accurately reflect our approach:an automated engine (the Kneedle algorithm) that operates within human-specified safety constraints (bottleneck layer retention).The primary value of our framework remains its ability to independently and automatically determine the optimal budget for the intermediate heterogeneous layers (representing over 90% of the model).
>
> We appreciate your comments as they help us clarify the practical considerations of MoE pruning.

---

> > ### Author Rebuttal · Reviewer_cWCS · 2026-04-01
> >
> > Thanks for the reply. I will maintain my positive score.

---

### Official Review · Reviewer_Pum5 · 2026-03-12

**Soundness:** 4
**Presentation:** 3
**Significance:** 3
**Originality:** 3
**Overall Recommendation:** 5
**Confidence:** 3

**Summary:**

The paper presents SCHUR-A*, a method for pruning experts in MoE models to reduce their memory footprint. Most pruning techniques rank experts individually, but this often fails because experts work together in complex ways. To solve this, the authors treat pruning as a "subset selection" problem, looking for the specific group of experts in each layer that minimizes the loss of information when the rest are removed.

The method builds upon the A* algorithm. This algorithm explores different combinations of experts to find the globally optimal group for each layer. To reduce search complexity, they use a "Schur-complement" math trick to create a tight upper bound. This bound allows the search to safely ignore bad combinations while guaranteeing it still finds the best possible subset.

To make the search fast enough for large models, the authors developed a Vectorized Gain Computation. Instead of checking experts one by one, which is slow, this scheme evaluates all possible candidates at once, speeding up the process significantly.

The method doesn't require the user to manually guess how many experts to keep for each layer: it uses an automated strategy called the Kneedle algorithm to find the "knee point" where adding more experts gives diminishing returns.

**Compliance With Llm Reviewing Policy:**

Affirmed.

**Final Justification:**

I think the paper is solid and my questions have been answered during the rebuttal so I maintain my "Accept". Thanks for a good paper!

**Key Questions For Authors:**

Is the Cholesky decomposition reliable in FP16 settings, as highly correlated experts can lead to ill-conditioned matrices that "crash" the math?

The testing is primarily on fine-grained MoEs; is the overhead of $A^*$ search worth it for simpler models with only 8 experts where greedy search might already be "good enough"?

**Limitations:**

yes

**Strengths And Weaknesses:**

### Strengths

Mathematical Rigor: The method is built on a solid theoretical foundation, specifically using the Schur-complement to derive a heuristic that guarantees the $A^*$ search finds a layer-wise global optimum.

Empirical results: The authors tested the method on models like Qwen3-30B and ERNIE-4.5, showing that it holds up even under heavy pruning (up to 50%) where other methods fail.

The paper includes a dedicated section on limitations, discussing numerical stability issues and the fact that layer-wise optimality doesn't always equal end-to-end model optimality.

The paper solves a real issue: it addresses the memory footprint of MoE models, which is a major hurdle for anyone trying to run these models.

By reducing search complexity from cubic to quadratic, the method makes global optimization feasible for models with many experts.

The use of the Kneedle algorithm means users don't have to manually tune the number of experts for their model architecture.

### Weaknesses

The performance is tied to the 512-sequence calibration set; if this data is biased or too narrow, the "optimal" experts chosen might not generalize to other domains.

While the authors mention 10 minutes of search time on an A100, the paper could be clearer about the total "wall-clock" time for an entire 30B model compared to simpler greedy methods.

Local vs. Global Optima: The method guarantees the best experts per layer, but it doesn't account for how layers interact with each other.

---

> ### Author Rebuttal · Authors · 2026-03-31
>
> We sincerely thank you for your positive assessment and for recognizing the mathematical rigor,efficiency,and practical significance of SCHUR-A*.We address the weaknesses and key questions below:
>
> 1.We address the potential ill-conditioning risk through two mechanisms.First,we apply Ridge Regularization ($\epsilon$-damping) as described in Eq. 8 & 12,which adds a small regularization term to the Gram matrix to improve its conditioning.Second,we observed that despite this regularization,the ill-conditioning issue still manifests in the first and last few layers,where experts are highly non-redundant and the Gram matrix remains near-singular.For these boundary layers,we adopted a straightforward solution: directly retaining all experts at full capacity,thereby bypassing the Cholesky decomposition entirely.This two-pronged design simultaneously addresses numerical instability and preserves the critical information flow at the model's input and output interfaces.
>
> 2.We agree that for simpler MoE architectures with a small number of experts (e.g., 8 experts in Mixtral 8x7B),greedy search or even exhaustive enumeration is not only feasible but may be preferable—the combinatorial space ($\binom{8}{k}$) is small enough that both approaches converge rapidly to the global optimum without the overhead of A* bookkeeping.However,SCHUR-A* targets a fundamentally different regime:state-of-the-art MoEs are rapidly scaling toward fine-grained designs with 64–128+ routed experts per layer.In these massive search spaces ($\binom{128}{k}$ easily exceeds $10^{20}$),greedy search is generally more prone to suboptimal decisions in these large combinatorial spaces—it cannot recover from early mistakes and systematically misses synergistic expert combinations.Exhaustive enumeration becomes computationally intractable.SCHUR-A* thus occupies the principled middle ground:it is unnecessary overhead for small-$n$ regimes,but the path to near-optimal solutions in large-$n$ fine-grained MoEs—precisely the architectures where compression matters most for deployment.
>
> 3.The choice of 512 sequences from C4 was intended as a more conservative and ample setting compared to established norms in LLM pruning.In fact, a large body of research has demonstrated that even 128 sequences (2048 tokens each) are sufficient to achieve stable reconstruction error for models as large as OPT-175B,with diminishing marginal returns for additional data.Given our focus on models in the 21B-30B range,512 sequences (4× the standard) provides a substantial margin for stability.Regarding domain and composition,we agree that the calibration distribution influences expert selection;we selected C4 to maintain the model's general intelligence.To directly address this concern,we have conducted a calibration sensitivity analysis as part of this rebuttal,comparing performance across different calibration set sizes (128 vs.256 vs.512 sequences).Results are shown in the table below：
> | Method | arc_c | arc_e | boolq | hellas | mmlu | obqa | rte | wino | avg |
> | :--- | :---: | :---: | :---: | :---: | :---: | :---: | :---: | :---: | :---: |
> | 128 | 0.491 | 0.717 | 0.869 | 0.758 | 0.627 | 0.434 | 0.801 | 0.683 | 0.673 |
> | 256 | 0.490 | 0.687 | 0.865 | 0.771 | 0.628 | 0.429 | 0.782 | 0.678 | 0.666 |
> | 512 | 0.491 | 0.662 | 0.866 | 0.760 | 0.623 | 0.452 | 0.740 | 0.699 | 0.661 |
>
> Conclusion:These results suggest that SCHUR-A* is robust across calibration sizes,and that larger calibration sets do not necessarily provide additional gains.We will revise the paper to clarify that 512 sequences were chosen as a conservative default rather than a heavily tuned optimum.
>
> 4.We would like to clarify that the search across all layers is executed in parallel.The reported "10 minutes" refers to the wall-clock time of the single most time-consuming layer;the vast majority of layers complete within 1–2 minutes.More importantly,we note that the dominant time cost in expert pruning—for both SCHUR-A and any greedy baseline—is not the search itself,but the shared prerequisite:forward inference over the calibration corpus to collect activation statistics and Gram matrices.This data collection phase must be performed once regardless of which search strategy is used.Once the calibration data is prepared,the search overhead is marginal by comparison.We will make this distinction clearer in the revised manuscript.
>
> 5.We agree that layer-wise optimality does not strictly guarantee end-to-end global optimality.Our study focuses on the layer-wise reconstruction objective,a widely-accepted surrogate in established post-training compression frameworks (e.g., GPTQ, SparseGPT, Wanda).We will further clarify this in our Limitations section and consider cross-layer joint optimization as a key direction for future work.

---

> > ### Author Rebuttal · Reviewer_Pum5 · 2026-04-01
> >
> > Thank you for your response, the 5 points you make address the questions I had about the paper.
> > One last element that might improve the paper further would be to run a comparison against an "oracle", i.e. exhaustively search across all possible variants of a (small) model, and compare against your layer-wise search.

---

### Official Review · Reviewer_cCEJ · 2026-03-13

**Soundness:** 2
**Presentation:** 3
**Significance:** 3
**Originality:** 3
**Overall Recommendation:** 4
**Confidence:** 3

**Summary:**

The paper proposes SCHUR-A*, a post-training pruning method for sparse MoE LLMs. The core idea is to reformulate layer-wise expert pruning as a reconstruction-driven subset selection problem and solve it using A* search guided by a novel admissible heuristic derived from the Schur complement of the Gram matrix. The method includes three key components: (1) the Schur-complement-based upper bound that enables efficient search-space pruning while guaranteeing layer-wise optimality, (2) a vectorized gain computation scheme using incremental Cholesky updates that reduces complexity from cubic to quadratic, and (3) an adaptive per-layer budget selection strategy using the Kneedle algorithm. The submission reports results mainly on Qwen3-30B-A3B, with additional appendix results on ERNIE-4.5-21B-A3B, and shows better accuracy and perplexity than ranking-based baselines and than a same-metric greedy variant.

**Compliance With Llm Reviewing Policy:**

Affirmed.

**Final Justification:**

I maintain my score as the rebuttal has addressed my concerns

**Key Questions For Authors:**

See weaknesses.

**Limitations:**

Yes

**Strengths And Weaknesses:**

## Strengths
1. The Schur-complement heuristic appears mathematically well aligned with the objective, and the decomposition leading to exact marginal gains is a useful derivation.

2. The ablation, which compares against a greedy method using the same gain metric, is good and clear. It helps isolate the value of global search rather than only the value of the score itself.

3. The proposed Vectorized Gain Computation via Cholesky updates is a strong technical contribution. By simultaneously evaluating candidates, it effectively reduces the computational complexity

## Weaknesses
1. Model diversity is limited. The paper evaluates on only two MoE architectures (Qwen3-30B-A3B and ERNIE-4.5-21B-A3B). More models should be included, like DeepSeek-V2 Lite or Moonlight-MoE.

2. The main empirical comparisons are not matched on pruning budget. SCHUR-A* uses adaptive pruning at about 42% on Qwen, while baselines are shown only at 25% and 50%. This makes the headline comparison not fair. A stronger evaluation would compare all methods at the same final retention ratio, and also compare adaptive-budget versions of baselines.

3.   The method relies on 512 sequences from C4 for calibration. There is no analysis of sensitivity to calibration set size, domain, or composition.

---

> ### Author Rebuttal · Authors · 2026-03-31
>
> Thank you for your constructive feedback and for acknowledging the mathematical alignment of our heuristic and the technical contribution of our vectorized computation scheme.We address the identified weaknesses below:
>
> 1.We agree that broader architectural coverage strengthens the paper.While our initial selection of Qwen3 (pure-specialization) and ERNIE-4.5 (shared-routed) was intended to represent the two dominant MoE paradigms,we appreciate the suggestion to include more models.We have now added evaluations on DeepSeek-V2-Lite to further confirm our method’s generalization across diverse architectures, as shown below:
> | Method | arc_c | arc_e | boolq | hellas | mmlu | obqa | rte | wino | avg |
> | :--- | :---: | :---: | :---: | :---: | :---: | :---: | :---: | :---: | :---: |
> | DeepSeek-V2-Lite | 0.464 | 0.742 | 0.801 | 0.778 | 0.554 | 0.446 | 0.606 | 0.708 | 0.637 |
> | SCHUR-A* | 0.304 | 0.516 | 0.514 | 0.588 | 0.264 | 0.358 | 0.487 | 0.552 | 0.448 |
> | kne-freq | 0.258 | 0.353 | 0.503 | 0.455 | 0.240 | 0.280 | 0.545 | 0.493 | 0.391 |
> | kne-EAN | 0.300 | 0.470 | 0.527 | 0.568 | 0.251 | 0.320 | 0.473 | 0.517 | 0.428 |
> | kne-REAP | 0.308 | 0.491 | 0.513 | 0.584 | 0.279 | 0.317 | 0.479 | 0.533 | 0.438 |
>
> Conclusion:The key takeaway is that SCHUR-A* remains the best-performing pruning method under the same compression setting, even on a third architecture outside the main paper. While absolute drops are larger on DeepSeek-V2-Lite than on Qwen3/ERNIE, this suggests that DeepSeek-V2-Lite is intrinsically more pruning-sensitive,rather than indicating a failure of the proposed search strategy.We will add this result and discussion in the revision.
>
> 2.We appreciate the suggestion for a more granular comparison.While the 42% retention for SCHUR-A* was an automated "knee-point" identified by the Kneedle algorithm to showcase end-to-end automation,we agree that a fixed-budget evaluation is essential for an "apples-to-apples" assessment.Accordingly,we have added: (1) a direct comparison at a fixed 42% budget for all methods, and (2) an adaptive evaluation where Kneedle is applied to all baselines for a fair,automated-budget comparison.Results are as follows:
>
> | Method | arc_c | arc_e | boolq | hellas | mmlu | obqa | rte | wino | avg |
> | :--- | :---: | :---: | :---: | :---: | :---: | :---: | :---: | :---: | :---: |
> | SCHUR-A* | 0.491 | 0.662 | 0.866 | 0.760 | 0.623 | 0.452 | 0.740 | 0.699 | 0.661 |
> | kne-freq | 0.380 | 0.555 | 0.832 | 0.674 | 0.563 | 0.382 | 0.610 | 0.656 | 0.581 |
> | kne-EAN | 0.418 | 0.554 | 0.839 | 0.685 | 0.569 | 0.377 | 0.591 | 0.668 | 0.588 |
> | kne-REAP | 0.473 | 0.633 | 0.846 | 0.736 | 0.602 | 0.408 | 0.755 | 0.683 | 0.642 |
> | 42%freq | 0.395 | 0.577 | 0.822 | 0.584 | 0.590 | 0.400 | 0.643 | 0.646 | 0.582 |
> | 42%EAN | 0.398 | 0.583 | 0.828 | 0.590 | 0.592 | 0.400 | 0.629 | 0.658 | 0.585 |
> | 42%REAP | 0.467 | 0.682 | 0.835 | 0.694 | 0.628 | 0.402 | 0.769 | 0.679 | 0.645 |
>
> Conclusion:SCHUR-A* consistently outperforms other baseline pruning criteria across multiple benchmarks.This confirms that our global optimization framework provides a more reliable foundation for maintaining model integrity during aggressive expert reduction compared to simple heuristic-based selection.
>
> 3.The choice of 512 sequences from C4 was intended as a more conservative and ample setting compared to established norms in LLM pruning.In fact, a large body of research has demonstrated that even 128 sequences (2048 tokens each) are sufficient to achieve stable reconstruction error for models as large as OPT-175B,with diminishing marginal returns for additional data.Given our focus on models in the 21B-30B range, 512 sequences (4× the standard) provides a substantial margin for stability.Regarding domain and composition,we agree that the calibration distribution influences expert selection; we selected C4 to maintain the model's general intelligence.To directly address this concern,we have conducted a calibration sensitivity analysis as part of this rebuttal,comparing performance across different calibration set sizes (128 vs.256 vs.512 sequences).Results are shown in the table below：
> | Method | arc_c | arc_e | boolq | hellas | mmlu | obqa | rte | wino | avg |
> | :--- | :---: | :---: | :---: | :---: | :---: | :---: | :---: | :---: | :---: |
> | 128 | 0.491 | 0.717 | 0.869 | 0.758 | 0.627 | 0.434 | 0.801 | 0.683 | 0.673 |
> | 256 | 0.490 | 0.687 | 0.865 | 0.771 | 0.628 | 0.429 | 0.782 | 0.678 | 0.666 |
> | 512 | 0.491 | 0.662 | 0.866 | 0.760 | 0.623 | 0.452 | 0.740 | 0.699 | 0.661 |
>
> Conclusion:These results suggest that SCHUR-A* is robust across calibration sizes,and that larger calibration sets do not necessarily provide additional gains.We will revise the paper to clarify that 512 sequences were chosen as a conservative default rather than a heavily tuned optimum.

---

> > ### Author Rebuttal · Reviewer_cCEJ · 2026-04-02
> >
> > Thanks for the substantive responses backed by new experimental evidence. I will maintain my score.

---

### Decision · Program_Chairs · 2026-04-30

**Decision:**

Accept (regular)

**Comment:**

This submission proposes SCHUR-A*, a reconstruction-driven pruning framework for sparse MoEs that casts layer-wise expert pruning as a constrained subset-selection problem and solves it using Schur-complement-guided A* search. I find the paper technically strong and sufficiently novel for acceptance. The main contribution is not merely another pruning score, but a principled combinatorial optimization framework with an admissible heuristic, exact marginal-gain derivation, and an efficient vectorized implementation that makes the approach practical on modern MoE models.

The empirical evidence is also reasonably convincing. In addition to outperforming ranking-based baselines, the paper includes a useful comparison against a greedy method using the same gain metric, which helps isolate the value of global search. The rebuttal materially strengthened the work through broader architectural validation, fixed-budget comparisons, and sensitivity analysis for calibration size. These additions address several important concerns and improve confidence that the method is robust within the scope claimed by the paper.

The limitations are real: performance degradation under aggressive pruning can still be large in absolute terms, calibration matters, and layer-wise optimality does not imply end-to-end optimality. However, these do not undermine the central contribution. The paper advances the methodology of post-training MoE pruning in a meaningful way and provides a rigorous upper-bound-oriented framework that others are likely to build on.

I therefore recommend accept. For revision, I encourage the authors to add more discussions and better acknowledge and discuss prior MoE pruning/compression work for instance, https://arxiv.org/pdf/2412.00069.